# Time to Effective Therapy Is an Important Determinant of Survival in Bloodstream Infections Caused by Vancomycin-Resistant *Enterococcus* spp

**DOI:** 10.3390/ijms231911925

**Published:** 2022-10-07

**Authors:** Alessandro Russo, Alice Picciarella, Roberta Russo, Gabriella d’Ettorre, Giancarlo Ceccarelli

**Affiliations:** 1Infectious and Tropical Disease Unit, Department of Medical and Surgical Sciences, “Magna Graecia” University of Catanzaro, 88100 Catanzaro, Italy; 2Internal Medicine Unit, Spoke Paola-Cetraro, ASP Cosenza, 87022 Cosenza, Italy; 3Internal Medicine Unit, Policlinico Casilino, 00169 Rome, Italy; 4Policlinico “Umberto I”, Department of Public Health and Infectious Diseases, “Sapienza” University, 00185 Rome, Italy

**Keywords:** bloodstream infections, enterococci, VRE, effective therapy, mortality

## Abstract

Enterococcal bloodstream infections (EBSI) caused by vancomycin-resistant enterococci (VRE) are associated with a significant rate of unfavorable outcomes. No definitive data have been reported about the association between delayed antibiotic therapy and mortality. In this prospective observational study in three large hospitals in Italy (from August 2016 to April 2021), all consecutive hospitalized patients with a confirmed diagnosis of hospital-acquired monomicrobial BSI caused by VRE—with no evidence of endocarditis—were analyzed. Cox regression analysis showed that risk factors independently associated with 30-day mortality were age (HR 2.98, CI95% 1.44–6.81, *p* = 0.002), chronic kidney disease (HR 5.21, CI95% 1.48–22.23, *p* = 0.001), oncologic disease (HR 2.81, CI95% 1.45–19.8, *p* = 0.005), and intensive care unit admission (HR 3.71, CI95% 2.23–7.99, *p* < 0.001). Conversely, early effective therapy was associated with survival (HR 0.32, CI95% 0.38–0.76, *p* < 0.001). The administration of early effective antibiotic therapy within 48 h from blood culture collection was associated with 30-day mortality rates lower than 33%. Time from blood culture collection to appropriate therapy was an independent predictor of 30-day mortality in patients with EBSI caused by VRE. Based on these data, clinicians should start effective antibiotic therapy as soon as possible, preferably within the first 48 h from blood culture collection. Treatment strategies allowing the early delivery of in vitro active antibiotics are urgently needed, especially in critically ill patients at risk of VRE bacteremia.

## 1. Introduction

Enterococcal bloodstream infections (EBSI) are associated with a significant rate of morbidity and mortality [1]. Moreover, treatment has become increasingly difficult due to emergence of multidrug-resistant strains [2], especially vancomycin-resistant enterococci (VRE) that have a predominance of *E. faecium* isolates [3].

VRE isolation is considered as an independent predictor of mortality among patients with EBSI [4], compared to vancomycin-susceptible infections. A delayed appropriate antibiotic therapy is considered an important determinant of an unfavorable outcome [5]. However, important questions regarding the true impact of delayed therapy on mortality are currently debated in patients with ESBI [6,7]. Many published studies have shown different results between time to appropriate therapy and outcomes for EBSI, as well as in determining a critical time period in which EBSI patients should receive appropriate therapy.

The aim of this study, therefore, is to evaluate the impact of an early effective antibiotic therapy on survival in a multicenter cohort of patients with EBSI caused by VRE.

## 2. Results

During the study period, 103 consecutive hospitalized patients with a diagnosis of monomicrobial EBSI, without evidence of endocarditis, were enrolled. Out of these, 79 (76.7%) were caused by *Enterococcus faecium* and 24 (23.3%) by *Enterococcus faecalis.* In accordance with local protocol for management of BSI, patients with positive blood cultures were evaluated by an infectious disease consultant team. All patients underwent echocardiography to exclude endocarditis.

Overall, the median age was 79.4 years (interquartile range [IQR], 55–95 years), the Charlson comorbidity index was recorded at 3.1 points (IQR, 1–6 points), and the time to effective therapy (based on in vitro isolate susceptibility) from blood culture collection was 3.12 days (IQR, 1–8 days). In regard to enterococcal isolates, nine (8.7%, all *Enterococcus faecium*) and two (1.9%, all *E. faecalis*) were recognized as daptomycin nonsusceptible and linezolid nonsusceptible, respectively. In regard to secondary BSI, the most frequent source of infection were: urinary tract infection (49%); CVC-related BSI (35%); intra-abdominal infections (13%); and skin and soft tissue infection (35). The most frequently used antibiotic regimens were: daptomycin plus ceftaroline (36.9%), only daptomycin (24.2%), only linezolid (18.4%), daptomycin plus ceftriaxone (9.7%), daptomycin plus ampicillin (8.7%), and daptomycin plus fosfomycin (2%). Fifty seven (55.3%) patients were treated with an early effective therapy (<48 h) from blood culture collection, while 46 (44.7%) patients were treated with a delayed effective therapy (>48 h) (see Table 1).

Seven (12.2%) patients treated with early effective therapy died at 30 days, compared to 21 (45.6%) patients treated with delayed effective therapy (*p* < 0.001) (Table 2).

Cox regression analysis showed that the risk factors independently associated with 30-day mortality were age (HR 2.98, CI95% 1.44–6.81, *p* = 0.002), chronic kidney disease (HR 5.21, CI95% 1.48–22.23, *p* = 0.001), oncologic disease (HR 2.81, CI95% 1.45–19.8, *p* = 0.005), and intensive care unit admission (HR 3.71, CI95% 2.23–7.99, *p* < 0.001). Conversely, early effective therapy was associated with survival (HR 0.32, CI95% 0.38–0.76, *p* < 0.001) (see Table 3).

The association between early effective therapy and the outcome in all of the study population was assessed using Cox regression analysis; further, the analysis considered initiation of effective antibiotic treatment as a time-varying covariate (see Table 4).

Kaplan–Meier curves for 30-day survival and rate of 30-day mortality with the start of an active targeted therapy are reported in Figure 1.

## 3. Discussion

This study highlights the relation between time to effective therapy and 30-day mortality in a large cohort of patients with EBSI caused by VRE isolates. As a point of interest, the time from blood culture collection to effective antibiotic therapy was independently associated with improved survival in these patients, even after adjustment for age, comorbidities, severity of illness, and therapy.

Specifically, our study demonstrated that the administration of early effective antibiotic therapy within 48 h from blood culture collection was associated with 30-day mortality rates lower than 33%. In this population, the median turnaround time from specimen collection to antimicrobial susceptibility testing was 3.12 days. Thus, these data underline the importance to deliver an effective therapy against VRE in patients with monomicrobial EBSI as soon as possible, and specifically within the first 48 h from the collection of blood cultures [8]. Based on these results, the rapid identification of patients with a high probability of BSI caused by VRE represents a great challenge for clinicians [9].

In this scenario, rapid diagnostic test represents a standard-of-care in management of BSI. Rapid identification of the *Enterococcus* species and the presence of vancomycin-resistant genes (VanA, VanB) has been shown to significantly reduce time to appropriate therapy and mortality of patients with hospital-onset EBSI. Moreover, the use of a rapid molecular diagnostic microarray assay to detect enterococcal species in blood cultures, has resulted in a mean time to appropriate therapy of approximately 24 h, which is accompanied by a reduction in hospital cost [10,11].

The observational nature and the small sample size are intrinsic study limitations. Additionally, we used blood culture collection time as the point from which to assess the timing of effective antibiotic therapy, which may introduce a bias in the determination of the timing of effective therapy. Regardless, the time from blood culture collection to an effective antibiotic therapy resulted as the main factor associated with 30-day mortality.

## 4. Materials and Methods

### 4.1. Study Design

We conducted a prospective observational study in 3 large hospitals (from 300 to 1200 beds) in Rome, Italy. From August 2016 to April 2021, all consecutive hospitalized patients with a confirmed diagnosis of hospital-acquired monomicrobial BSI caused by VRE, with no evidence of endocarditis, were analyzed.

### 4.2. Patient Consent Statement

The local ethics committees of each participating center approved the study protocol for observational analyses. The informed consent was obtained from each patient included in the study, and the study was conducted in accordance with the principles of the Declaration of Helsinki.

### 4.3. Data Collection and Definitions

Patient data from medical records, computerized hospital databases, and/or clinical charts were prospectively collected using a standard form. They included demographics, clinical and laboratory findings, baseline comorbidities, radiological findings, relapse of infection, microbiological data, the Charlson comorbidity index, source of infection, development of septic shock, duration of hospital stay, duration of definitive antibiotic therapy, clinical treatment failure, and 30-day mortality.

The following definitions were established prior to data analysis: *relapse*—a new diagnosis of EBSI caused by the same organism after clinical and microbiological resolution of a previous episode of treated EBSI; *persistent infection*—patients treated with an active in vitro antibiotic regimen without resolution of EBSI; *clinical treatment failure*—lack of response to the definitive antimicrobial regimen, as reflected by the presence of any of the following: ongoing fever, leukocytosis, or other clinical signs of infection that could not be attributed to causes other than EBSI; *30-day mortality*—death from any cause within the 30 days following diagnosis of EBSI.

### 4.4. Microbiological Identification

The identification of etiology was based in accordance with local laboratory techniques. From positive cultures, Gram staining and a rapid identification protocol were adopted. The bacterial pellet obtained directly from positive cultures was used for MALDI-TOF MS (Bruker Daltonics, Billerica, MA, USA) identification and for molecular analysis. The SensiTitre™ system (Thermo Fisher Scientific, Waltham, MA, USA) or the Vitek 2 automated system (bioMérieux, Marcy l’Etoile, France) were used for isolate identification and antimicrobial susceptibility testing. Minimum inhibitory concentrations (MICs) were established according to the European Committee on Antimicrobial Susceptibility Testing (EUCAST) breakpoints [12,13]. A multiplex PCR, combined with array detection, using an automated closed system that isolates, amplifies, and detects nucleic acid for multiple causative pathogens within a single specimen in one step (FilmArray, BioFire Diagnostics, BioMérieux, Salt Lake City, UT, USA) was used, from 2018, to identify the possible organisms and the corresponding bacterial resistance genes within 60–90 min.

### 4.5. Antimicrobial Treatment Evaluation

Depending on the number of drugs used (1 or >1), treatment regimens were classified either as monotherapy or combination therapy. Antibiotic therapy was defined with accordance to in vitro isolate susceptibility. Drugs in effective therapy must have been administered for at least 50% of the total duration of therapy. Time to effective therapy was the period between the blood culture collection and initial effective therapy. Early effective therapy was defined as the therapy administered in the first 48 h from blood culture collection, while delayed effective therapy was defined as the therapy administered after 48 h.

### 4.6. Primary Endpoints and Statistical Analysis

The primary endpoint was the analysis of risk factors associated with 30-day mortality. The differences between groups were assessed with the chi-square test or Fisher exact test (for categorical variables) and the two-tailed t test or Mann–Whitney test (for continuous variables), as appropriate. Univariate and multivariate Cox regression analyses were used to determine the effects of different variables on 30-day mortality in the study population. To avoid immortal time bias, the association between effective therapy and mortality was assessed with multivariable logistic regression analysis at pre-specified landmark times (1 and 48 h). Possible confounding factors and interactions were weighted during analysis with a backward stepwise selection and the consideration of *p* ≤ 0.05 for all variables in order to determine the effects of all anamnestic, clinical, and therapeutic variables on outcomes. We used clinical reasoning to identify confounding factors a priori. All reported *p*-values are two-tailed, Wald confidence intervals and hazard ratios were computed based on estimated standard errors. Survival curves for time-to-event variables, constructed using Kaplan–Meier estimates, were based on all available data and were compared with the use of the log-rank test. The association between effective therapy and outcomes was assessed using Cox regression analysis with cause-specific hazard functions that considered the initiation of effective antibiotic treatment as a time-varying covariate. All analyses were conducted with the SPSS software package (version 22.0, SPSS Inc., Chicago, IL, USA).

## 5. Conclusions

In conclusion, our study shows that time from blood culture collection to appropriate therapy is an independent predictor of 30-day mortality in patients with EBSI caused by VRE. Of importance, EUCAST has never published breakpoints for daptomycin against enterococci, citing insufficient evidence [13]. Therefore, real data from clinical practice can help physicians to adequately manage this difficult-to-treat infection. As such, treatment strategies allowing the early delivery of in vitro active antibiotics are urgently needed, especially in critically ill patients at risk of VRE bacteremia.

## Figures and Tables

**Figure 1 ijms-23-11925-f001:**
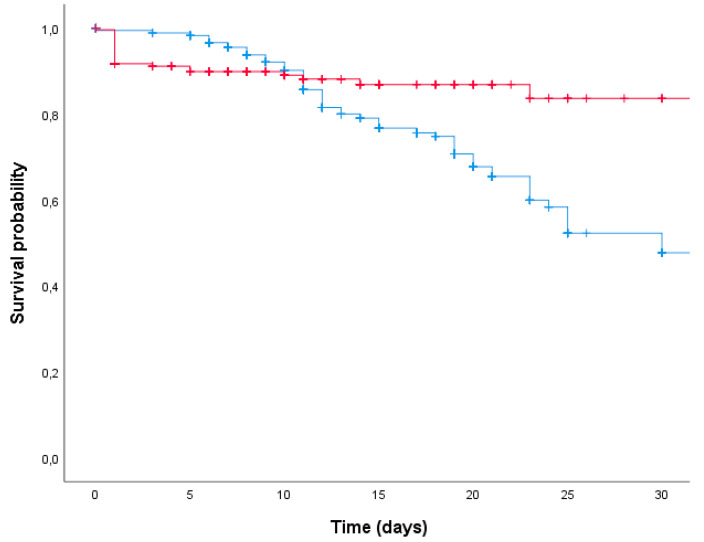
Kaplan–Meier curves on 30-day survival of patients when treated with early effective therapy (red line) or delayed effective therapy (blue line) (*p* < 0.001).

**Table 1 ijms-23-11925-t001:** Comparison between patients treated with early effective or delayed effective therapy.

Variables	Early Effective Therapy *n* = 57 (%)	Delayed Effective Therapy *n* = 46 (%)	*p*-Value
**Anamnestic factors**			
Age (years), median (IQR 25–75)	78.9 (53–93)	79.5 (56–96)	0.19
Male Sex	25 (43.9)	22 (47.8)	0.68
Previous VRE infection	3 (5.2)	1 (2.1)	0.41
Previous antibiotic therapy (30 days)	44 (77.2)	33 (71.7)	0.52
**Etiology and type of** **infection**			
*Enterococcus faecalis*	14 (24.5)	10 (21.7)	0.73
*Enterococcus faecium*	43 (75.4)	36 (78.2)	0.73
Secondary BSI	38 (66.6)	30 (65.2)	0.82
**Comorbidities**			
Charlson comorbidity index, median (IQR 25–75%)	3 (1–5)	3 (1–6)	0.56
Cardiovascular disease	32 (56.1)	26 (56.5)	0.96
Obesity	18 (31.5)	11 (23.9)	0.38
Diabetes mellitus type 2	21 (36.8)	15 (32.6)	0.65
Chronic kidney disease	11 (19.3)	7 (15.2)	0.58
Chronic obstructive pulmonary disease	31 (54.4)	23 (50)	0.65
Oncologic disease	19 (33.3)	11 (23.9)	0.29
Neurologic disorders	9 (15.8)	4 (8.7)	0.28
**Laboratory findings**			
Pitt score, median (IQR 25–75%)	8 (2–12)	8 (2–12)	1.0
Procalcitonin (ng/mL), median (IQR 25–75%)	2.5 (0.5–4)	2.5 (0.5–5)	0.65
**Antibiotic regimens**			
Daptomycin nonsusceptible	6 (10.5)	3 (5)	0.87
Linezolid nonsusceptible	2 (3.5)	–	0.19
Daptomycin plus ceftaroline	20 (35)	18 (39.1)	0.67
Daptomycin alone	14 (24.5)	11 (23.9)	0.93
Linezolid alone	11 (19.3)	8 (17.4)	0.8
Daptomycin plus ceftriaxone	6 (10.5)	4 (8.7)	0.75
Daptomycin plus ampicillin	5 (8.7)	4 (8.7)	0.98
Daptomycin plus fosfomycin	1 (1.7)	1 (2.1)	0.87
**Outcomes**			
ICU admission	8 (14)	26 (56.5)	**<0.001**
Adequate source control of infection	12 (21)	8 (17.4)	0.64
Length of ICU stay (days), median (IQR 25–75%)	13 (8–30)	19 (6–35)	**0.002**
Length of hospitalization (days), median (IQR 25–75%)	22 (7–41)	31 (6–49)	**0.004**
Septic shock	11 (19.3)	7 (15.2)	0.58
30-day mortality	7 (12.2)	21 (45.6)	**<0.001**

Legend. VRE: vancomycin-resistant enterococci; BSI: bloodstream infection; ICU: intensive care unit.

**Table 2 ijms-23-11925-t002:** Comparison between survivor and non-survivor patients.

Variables	Survivors *n* = 75 (%)	Non-Survivors *n* = 28 (%)	*p*-Value
**Anamnestic factors**			
Age (years), median (IQR 25–75%)	72.9 (52–83)	86 (66–95)	**<0.001**
Male Sex	37 (49.3)	10 (35.7)	0.21
Previous VRE infection	2 (4)	2 (7.1)	0.29
Previous antibiotic therapy (30 days)	54 (72)	23 (82.1)	0.29
**Etiology and type of** **infection**			
*Enterococcus faecalis*	18 (24)	6 (21.4)	0.78
*Enterococcus faecium*	57 (76)	22 (78.5)	0.78
Secondary BSI	48 (64)	20 (71.4)	0.64
**Comorbidities**			
Charlson comorbidity index, median (IQR 25–75%)	2.5 (1–3)	4 (2–6)	**<0.001**
Cardiovascular disease	38 (50.6)	20 (71.4)	0.06
Obesity	21 (28)	8 (28.5)	0.95
Diabetes mellitus type 2	26 (34.6)	10 (35.7)	0.92
Chronic kidney disease	7 (9.3)	11 (39.2)	**<0.001**
Chronic obstructive pulmonary disease	32 (42.6)	22 (78.5)	**0.001**
Oncologic disease	9 (12)	21 (75)	**<0.001**
Neurologic disorders	12 (16)	1 (3.5)	0.12
**Laboratory findings**			
Pitt score, median (IQR 25–75%)	7 (2–10)	9 (3–12)	**0.005**
Procalcitonin (ng/mL), median (IQR 25–75%)	2.5 (0.5–4)	2.5 (1–5)	0.43
**Antibiotic regimens**			
Daptomycin nonsusceptible	5 (6.6)	4 (14.2)	0.22
Linezolid nonsusceptible	1 (1.3)	1 (3.5)	0.46
Daptomycin plus ceftaroline	29 (38.6)	9 (32.1)	0.54
Daptomycin alone	17 (22.6)	8 (28.5)	0.53
Linezolid alone	14 (18.6)	5 (17.8)	0.92
Daptomycin plus ceftriaxone	7 (9.3)	3 (10.7)	0.83
Daptomycin plus ampicillin	7 (9.3)	2 (7.1)	0.72
Daptomycin plus fosfomycin	1 (1.3)	1 (3.5)	0.46
**Outcomes**			
ICU admission	12 (16)	22 (78.5)	**<0.001**
Time to effective therapy	2.5 (1–4)	4 (2–8)	**0.001**
Adequate source control of infection	15 (20)	5 (17.8)	0.8
Length of ICU stay (days), median (IQR 25–75%)	16 (7–31)	17 (7–36)	0.33
Length of hospitalization (days), median (IQR 25–75%)	26 (7–39)	30 (7–43)	0.08
Early effective therapy	50 (66.6)	7 (25)	**<0.001**
Delayed effective therapy	25 (33.3)	21 (75)	**<0.001**
Septic shock	7 (9.3)	11 (39.2)	**<0.001**

Legend. VRE: vancomycin-resistant enterococci; BSI: bloodstream infection; ICU: intensive care unit.

**Table 3 ijms-23-11925-t003:** Cox regression analysis on risk factors associated with 30-day mortality in all of study population.

Variables	Hazard Ratio	CI95%-Lower	CI95%-Upper	*p*-Value
Age	2.98	1.44	6.81	0.002
Oncologic disease	2.81	1.45	19.8	0.005
Chronic kidney disease	5.21	1.48	22.23	0.001
ICU admission	3.71	2.23	7.99	<0.001
Early effective therapy	0.32	0.38	0.76	<0.001

Legend. ICU: intensive care unit.

**Table 4 ijms-23-11925-t004:** Association between early effective antibiotic therapy and outcomes in all study patients.

Variables	Hazard Ratio	*p*-Value
ICU admission	0.12	<0.001
Septic shock	0.89	0.12
Length of ICU stay	0.22	0.001
Length of hospitalization	0.78	0.07
30-day mortality	0.08	<0.001

## Data Availability

Data are available from a.russo@unicz.it.

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
