# Peer review of "Time to Effective Therapy Is an Important Determinant of Survival in Bloodstream Infections Caused by Vancomycin-Resistant Enterococcus spp"

_ijms, 2022, doi:10.3390/ijms231911925_

Round 1
Reviewer 1 Report
This observational study investigates the effect of delayed antibiotic therapy on the survival of patients with Enterococcal bloodstream infections caused by VRE. The topic is of importance considering the high mortality caused by such infections and the difficulties to obtain results of antibiotic susceptibility testing of the bacteria rapidly. The study is well designed, and the article is well written. I have one major concern regarding the treatment of censored cases. Please find below my questions and suggestions.
Lines 22-24: You must mean delayed therapy after 48h instead of “early effective antibiotic therapy within 48 hours” and also it was 33,3% survival rate if I understand what you are referring too, so above 33%.
Line 41: mortality in patients with EBSI
Lines 42-45: Please reformulate this sentence.
Line 56: It would be useful to have a clear definition of what you intend by “effective therapy” early in the text. Do you mean therapy with an antibiotic to which the infecting bacteria are shown to not be resistant to?
Table 2: Why is there no p-value for neurologic disorders?
Figure 2: Could you improve the legend by replacing 0 and 1 by the actual condition represented? Additionally, in this figure, we can see that there were at least 49 censored cases. Please describe how you treated these cases for the other results presented.
Lines 96-98: Same issue as on lines 22-24
Line 146: in accordance with
Lines 161-166: The definitions are not expressed clearly. Additionally, you give the definition of some things that you never mention in the rest of the text e.g. “initial antibiotic therapy”
Lines 173-177: Which model was used for the cox regression in the end? Which variables were included?
Line 186-188: Theses sentences lack context and are a bit out of place in your conclusion.
Author Response
Reviewer 1
This observational study investigates the effect of delayed antibiotic therapy on the survival of patients with Enterococcal bloodstream infections caused by VRE. The topic is of importance considering the high mortality caused by such infections and the difficulties to obtain results of antibiotic susceptibility testing of the bacteria rapidly. The study is well designed, and the article is well written. I have one major concern regarding the treatment of censored cases. Please find below my questions and suggestions.
- dear reviewer, thank you very much for all your comments. We think that now the quality of the manuscript is improved. We tried to address all your suggestions.
Lines 22-24: You must mean delayed therapy after 48h instead of “early effective antibiotic therapy within 48 hours” and also it was 33,3% survival rate if I understand what you are referring too, so above 33%.
Line 41: mortality in patients with EBSI
Lines 42-45: Please reformulate this sentence.
Line 56: It would be useful to have a clear definition of what you intend by “effective therapy” early in the text. Do you mean therapy with an antibiotic to which the infecting bacteria are shown to not be resistant to?
Table 2: Why is there no p-value for neurologic disorders?
R: we modified manuscript as required.
Figure 2: Could you improve the legend by replacing 0 and 1 by the actual condition represented? Additionally, in this figure, we can see that there were at least 49 censored cases. Please describe how you treated these cases for the other results presented.
R: we modified Figure as required. About censored cases we recorded patients that not completed follow-period of 28 days for the main outcome. Censored were those patients who were no developed an event or not dead from ESBI.
Lines 96-98: Same issue as on lines 22-24
Line 146: in accordance with
Lines 161-166: The definitions are not expressed clearly. Additionally, you give the definition of some things that you never mention in the rest of the text e.g. “initial antibiotic therapy”
R: we modified manuscript as required.
Lines 173-177: Which model was used for the cox regression in the end? Which variables were included?
R: according also with comments of reviewer 2 we modified Cox regression analysis (see Methods section).
Line 186-188: Theses sentences lack context and are a bit out of place in your conclusion.
R: we modified manuscript as required.

Reviewer 2 Report
This is a multicenter prospective cohort study of patients with hospital-onset VRE bacteremia (without endocarditis). Among 103 patients, 57 patients received early effective therapy (within 48 hours from positive BCx) and 46 patients received delayed effective therapy. They found early effective therapy was independently associated with lower 30-day mortality (HR 0.32).
It is biologically plausible that early appropriate treatment improves outcome. Previous studies also demonstrated that shorter time to appropriate treatment was associated with improved outcome (Clinical Infectious Diseases® 2016;62(10):1242–50; Infect Drug Resist. 2022 Mar 1;15:723-734.) for patients with enterococcal bacteremia, including VRE. This study’s finding is not a brand-new one, but add to the existing literature. The study was fairly conducted with some questions in methodology, and well written.
My specific comments are below.
1. Title – Time to effective therapy is the most important determinant…. How the authors can say it is “the most” important determinant over other factors? I would suggest removing “the most” unless there is a convincing reason.
2. Introduction, lines 39-40. Authors claimed there is no definitive data about the association between delayed antibiotic therapy and mortality in EBSI. As stated in my general comments, at least several observational studies which assessed the relationship. Although they are not definitive because of observational design, I do not think RCT will be conducted for this due to ethical concern. So, I do not think it fair to claim there is no definitive data yet. I suggest to modify this part.
3. Results, table 1 and 2. I would recommend adding headings to variables such as comorbidities, drug susceptibility, treatment regimen, outcome etc. In addition, would be good to include the contents of secondary BSI as mortality would be different from UTI, intra-abdominal, CRBSI etc.(if data is available)
4. Results, figure 1. I do not think it is needed, as the information is already shown in table 1.
5. Method, figure 2 and table 3. I have a couple of questions about their statistical analysis. First, it seems authors use the time of positive blood culture as index time (time 0), and categorize patients who had effective treatment within 48 hours as early group, and effective treatment after 48 hours as delayed group. It means they did not know a patient was categorized in which group at time 0. It could lead to significant bias (immortal time bias). If there was no patient who died before getting effective treatment in this study, immortal time bias may not be an issue. But I could not find if it is true. Alternatively, if time 0 is the beginning of effective therapy, there is no immortal time bias. Consider modifying the analysis. At least, authors need to explain how they addressed immortal time bias in the method section.
6. Method, figure 2 and table 3. Second question is about the Cox regression analysis. It seems they used the Cox proportional regression given only one hazard ratio per variable was provided in table 3. But looking at K-M curve, obviously proportional hazard assumption is violated (two survival curves crossed). In that case, Cox proportional regression should not be used because the hazard of mortality changes over time. I suggest using the method to include time-varying coefficients to calculate time-varying hazard ratios. (BMC Med Res Methodol. 2010 Mar 16;10:20.; Statistics in medicine, 2011, Vol.30 (3), p.250-259)
7. Discussion, line 113-116. I did not understand what authors wanted to say in this sentence. Probably need to rephrase.
Author Response
Reviewer 2
This is a multicenter prospective cohort study of patients with hospital-onset VRE bacteremia (without endocarditis). Among 103 patients, 57 patients received early effective therapy (within 48 hours from positive BCx) and 46 patients received delayed effective therapy. They found early effective therapy was independently associated with lower 30-day mortality (HR 0.32).
It is biologically plausible that early appropriate treatment improves outcome. Previous studies also demonstrated that shorter time to appropriate treatment was associated with improved outcome (Clinical Infectious Diseases® 2016;62(10):1242–50; Infect Drug Resist. 2022 Mar 1;15:723-734.) for patients with enterococcal bacteremia, including VRE. This study’s finding is not a brand-new one, but add to the existing literature. The study was fairly conducted with some questions in methodology, and well written.
- dear reviewer, we are grateful for all your comments. We think that specification about methods improved quality of the manuscript.
My specific comments are below.
- Title – Time to effective therapy is the most important determinant…. How the authors can say it is “the most” important determinant over other factors? I would suggest removing “the most” unless there is a convincing reason.
- Introduction, lines 39-40. Authors claimed there is no definitive data about the association between delayed antibiotic therapy and mortality in EBSI. As stated in my general comments, at least several observational studies which assessed the relationship. Although they are not definitive because of observational design, I do not think RCT will be conducted for this due to ethical concern. So, I do not think it fair to claim there is no definitive data yet. I suggest to modify this part.
- Results, table 1 and 2. I would recommend adding headings to variables such as comorbidities, drug susceptibility, treatment regimen, outcome etc. In addition, would be good to include the contents of secondary BSI as mortality would be different from UTI, intra-abdominal, CRBSI etc.(if data is available)
- Results, figure 1. I do not think it is needed, as the information is already shown in table 1.
R: we modified manuscript as required.
- Method, figure 2 and table 3. I have a couple of questions about their statistical analysis. First, it seems authors use the time of positive blood culture as index time (time 0), and categorize patients who had effective treatment within 48 hours as early group, and effective treatment after 48 hours as delayed group. It means they did not know a patient was categorized in which group at time 0. It could lead to significant bias (immortal time bias). If there was no patient who died before getting effective treatment in this study, immortal time bias may not be an issue. But I could not find if it is true. Alternatively, if time 0 is the beginning of effective therapy, there is no immortal time bias. Consider modifying the analysis. At least, authors need to explain how they addressed immortal time bias in the method section.
R: thanks for this important observation. To reduce the bias about immortal time we considered time 0 as the beginning of effective therapy; moreover we decided to include only patients with drugs in effective therapy administered for at least 50% of the total duration of therapy. According also with your other comments below we modified statistical analysis and reported Table 4.
- Method, figure 2 and table 3. Second question is about the Cox regression analysis. It seems they used the Cox proportional regression given only one hazard ratio per variable was provided in table 3. But looking at K-M curve, obviously proportional hazard assumption is violated (two survival curves crossed). In that case, Cox proportional regression should not be used because the hazard of mortality changes over time. I suggest using the method to include time-varying coefficients to calculate time-varying hazard ratios. (BMC Med Res Methodol. 2010 Mar 16;10:20.; Statistics in medicine, 2011, Vol.30 (3), p.250-259)
R: we modified analysis as required. See methods section and Table 4.
- Discussion, line 113-116. I did not understand what authors wanted to say in this sentence. Probably need to rephrase.
R: we modified manuscript as required.

Round 2
Reviewer 2 Report
This revised manuscript addressed my questions appropriately.